# An Interference-Resistant and Low-Consumption Lip Recognition Method

**Junwei Jia [1], Zhilu Wang [2], Lianghui Xu [1], Jiajia Dai [1], Mingyi Gu [1] and Jing Huang [1],***

[1] School of Information and Electronic Engineering, Zhejiang Gongshang University (ZJSU), Hangzhou 310018, China
[2] College of Mechanical and Electrical Engineering, Hohai University (HHU), Changzhou 213022, China
* Correspondence: jhuang@mail.zjgsu.edu.cn

**Abstract:** Lip movements contain essential linguistic information. It is an important medium for studying the content of the dialogue. At present, there are many studies on how to improve the accuracy of lip language recognition models. However, there are few studies on the robustness and generalization performance of the model under various disturbances. Specific experiments show that the current state-of-the-art lip recognition model significantly drops in accuracy when disturbed and is particularly sensitive to adversarial examples. This paper substantially alleviates this problem by using Mixup training. Taking the model subjected to negative attacks generated by FGSM as an example, the model in this paper achieves 85.0% and 40.2% accuracy on the English dataset LRW and the Mandarin dataset LRW-1000, respectively. The correct recognition rates are improved by 9.8% and 8.3%, compared with the current advanced lip recognition models. The positive impact of Mixup training on the robustness and generalization of lip recognition models is demonstrated. In addition, the performance of the lip recognition classification model depends more on the training parameters, which increase the computational cost. The InvNet-18 network in this paper reduces the consumption of GPU resources and the training time while improving the model accuracy. Compared with the standard ResNet-18 network used in mainstream lip recognition models, the InvNet-18 network in this paper has more than three times lower GPU consumption and 32% fewer parameters. After detailed analysis and comparison in various aspects, it is demonstrated that the model in this paper can effectively improve the model's anti-interference ability and reduce training resource consumption. At the same time, the accuracy is comparable with the current state-of-the-art results.

**Keywords:** lip recognition; visual speech recognition; data enhancement; inverse convolutional neural network

## 1. Introduction

Lip recognition is also called visual speech recognition (VSR). It analyzes the dynamic changes of the lips. The aim is to recognize the speech content in the video. This task involves natural language processing, image classification, speech processing, and pattern recognition. In recent years, there have been many applications of lip recognition in real life such as in vivo detection [1], improved hearing aids [2], etc., with broad application prospects.

The lip recognition model consists of two steps. The first is to extract the visual features of the lips. The second is categorization. The extracted visual features should contain sufficient representative information and robustness [3]. The traditional extraction method is manual annotation. Such practices only ensure that the downstream task can be classified and recognized without considering the effectiveness of the acquired features. Therefore, the recognition accuracy is low. Although, there are corresponding methods [4,5] to solve this problem. These methods rely on manual design, and the design process is complex. Visual features obtained using manual annotation do not meet human expectations, and researchers have begun to seek more effective visual elements. Deep learning techniques

have been widely used in image detection in recent years. Deep learning techniques can automatically extract features from datasets, eliminating the hassle of manual extraction. Deep learning networks are flexible and can process large amounts of data. Recently, lip recognition based on deep learning has gradually become a research hotspot. The basic framework of deep-learning-based lip recognition is shown in Figure 1.

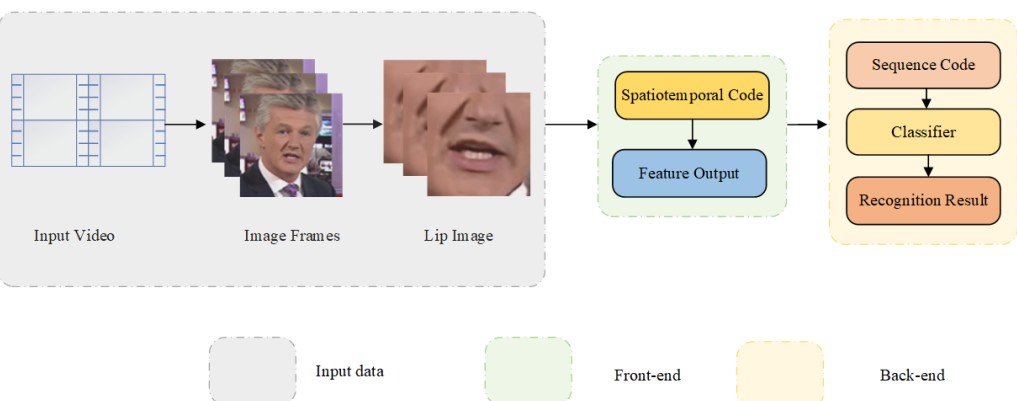

**Figure 1.** Deep-learning-based lip recognition framework.

Lip datasets are susceptible to adversarial examples. Achieving high-accuracy lip recognition under the interference of adversarial examples is one of the objectives of this paper. Meanwhile, most deep-learning-based lip recognition models have a problem: too many model parameters due to the stacking of neural networks. These problems lead to a lot of GPU resource and time costs; this paper also gives methods to reduce GPU resource use and training time.

This paper summarizes the various types of deep-learning-based lip recognition models in Section 2. Then, we propose the baseline model in this paper based on the current advanced lip recognition models in Section 3. In Section 4, we offer an improved model based on the baseline model. We design a model based on Mixup and InvNet-18. Mixup has significantly enhanced the robustness and generalization of neural network architectures [6,7]. However, no research has yet demonstrated that Mixup can be applied to lip recognition models. This paper innovatively introduces Mixup training into a lip recognition model disturbed by adversarial examples. In Section 4.1, we demonstrate through detailed anti-interference experiments that current state-of-the-art lip recognition models have poor robustness and generalization against adversarial examples. Mixup training significantly improved anti-interference ability, robustness, and generalization performance. In Section 4.2, it is experimentally demonstrated that the InvNet-18 network in this paper can effectively reduce model parameters while maintaining accuracy, thus saving GPU resources and reducing training time. The InvNet-18 is an efficient and low-consumption deep neural network. In Section 5, on the datasets LRW [8] and LRW-1000 [9], we compare the model in this paper with other lip recognition models. It is demonstrated that the model's accuracy in this paper is comparable with the current state-of-the-art results in the case of interference resistance and low consumption.

## 2. Related Work

In this section, we derive the advantages and disadvantages of these models by analyzing lip recognition models. This step is essential because it helps us improve the lip recognition model.

Jinlin Ma et al. [10] classified lip recognition models into the following categories based on the differences in visual feature extraction methods: two-dimensional convolutional neural network (2D CNN), three-dimensional convolutional neural network (3D CNN), and 3D + 2D CNN. The specific classification of lip recognition is shown in Figure 2.

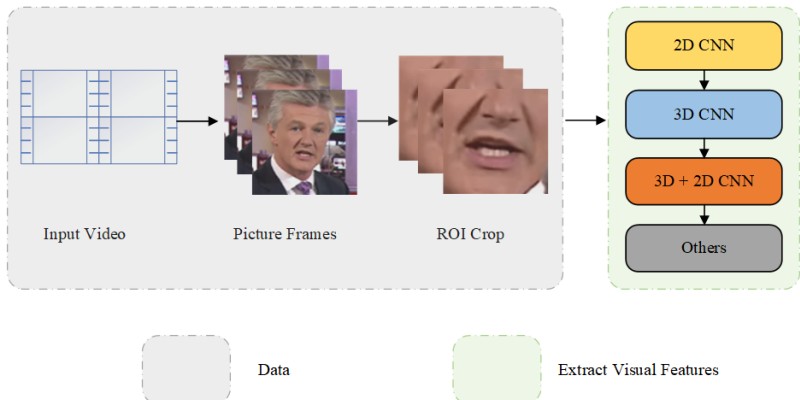

**Figure 2.** Visual feature extraction based on deep learning.

The flow chart of the 2D CNN-based lip recognition model is shown in Figure 3a. First, we used CNN to extract the visual features of lips. The CNN model consists of six convolution layers (convolution + non-linear activation + top pooling layer) and one whole connection layer. The CNN is trained using a combination of lip images and phoneme labels, and the output is used as visual features for lip recognition. The hidden Markov Model and Gaussian Mixed Observation Model were used in the back-end. This method solves the problem where the feature cannot be extracted automatically and the model cannot process the variable length sequence. Garg et al. [11] further improved the lip language model. They used VGGNet for variable length sequences. The image sequence is stitched into a single image as the model's input. The back-end uses Long Short-Term Memory (LSTM) to extract time information. The model performs well but faces two problems: how to obtain more visual features and how to reduce the computation of the model. Due to the limitations of a single model, Noda et al. [12] increased audio input as a model. They studied the correlation between speech and visual features in unlabeled visual speech recognition. They used depth encoding to extract audio features. Then, a multistream hidden Markov model was introduced to integrate the two-stream feature information to obtain the feature information. The model adaptively switches two channels' input features. However, the automatic weight selection is not implemented, which makes it challenging to use in practical applications.

In contrast, Lee et al. [13] suggested that multiview images can increase visual feature information. They used multiview images as the model's input. The front-end module uses stacked convolution layers to extract multiscale visual features, while the back-end module uses LSTM. The 2D CNN can only process single-frame images and is weak for continuous-frame images.

The flow chart of the 3D CNN-based lip recognition model is shown in Figure 3b. LipNet [14] was the first lip recognition model to introduce 3D CNN technology. The model takes a T-frame RGB image sequence as input. It is fed into a convolution network consisting of three layers of three-dimensional convolution. Each convolution network has a maximum pool layer behind it. The back-end network is further aggregated by a three-tier Bi-Gated Recurrent Unit (Bi-GRU), which aggregates the extracted features. Finally, the Connectionist Temporal Classification (CTC) is used for analysis loss, but CTC has its drawbacks: it requires input sequences to be larger than output sequences. Fung et al.'s front-end module [15] uses the same structure. However, differently, they used 83D convolution as a visual feature extractor. Although the model has good results, increased network depth easily hinders the flow of gradient information. Xu et al. [16] proposed an LCANet video encoder network for gradient information backflow. Send the input video to the overlaid 3D CNN. Using the two-layer Highway Network added to the 3D CNN solved the problem of gradient information return in the deep network. LCANet feeds the encoding information into an attention mechanism network to capture data in context. This attention mechanism weakens the constraint of the conditional independence hypothesis

on CTC. The attention mechanism improves the modeling ability of lip language models too. For large video datasets, deep 3D CNN improves classification accuracy. For example, Weng et al. [17] used the I3D dual-stream module as the front-end module. They used gray video frames and light streams as input to the front-end module. The model stitched the features of the two branches together. The back-end module used LSTM. Experiments have shown that light streams can obtain more visual elements when dealing with large-scale datasets. I3D also improves model recognition accuracy. Pratham et al. [18] used the SpotFast network as the visual feature extraction network to enhance recognition accuracy. A transformer learning sequence was used in the back-end model. Memory Enhanced networks (MEN) can effectively increase the capacity of the neural networks without increasing the computational amount. Experiments show that the performance of deep 3D CNN is further improved over the I3D network. With the increase in network layers, the 3D CNN model has two disadvantages: excessive parameters and high storage cost. Our model reduces the parameters through the InvNet-18 network, improves efficiency, and reduces the consumption of GPU resources. The main content and experiment are in Section 4.

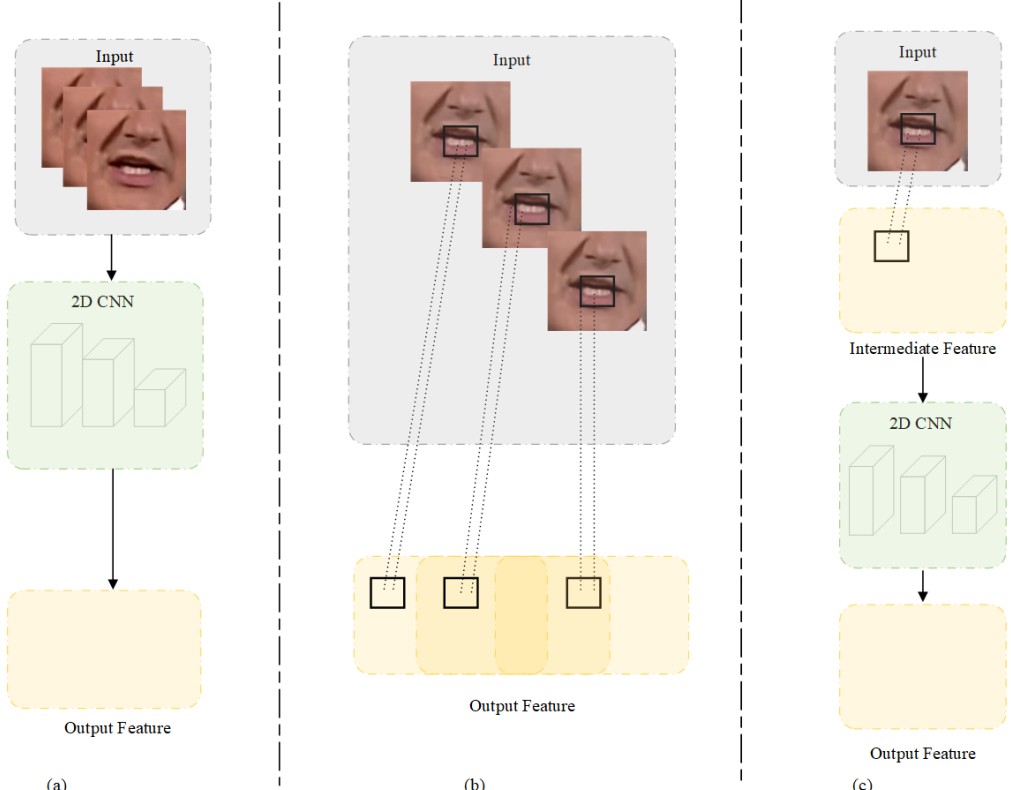

**Figure 3.** Visual feature extraction structure base CNN. (**a**) Base 2D CNN lip recognition models. (**b**) Base 3D CNN lip recognition models. (**c**) Base 3D + 2D CNN lip recognition models.

The most advanced approach is the lip recognition model based on the combination of 3D CNN and 2D CNN. The flow chart of the 3D + 2D CNN-based lip recognition model is shown in Figure 3c. There are two ways to combine 2D CNN with 3D CNN: the first method corrects the first layer of 2D CNN convolution to 3D CNN. A layer of 3D CNN captures the model's space-time information between successive frames. Then, the model connects to Depth 2D CNN to extract lip features. The second method is where the model uses a shallow 3D CNN to preprocess the video frame. Then, the model connects the standard deep 2D CNN. For the first method, Stafylakis [19] and Feng et al. [20] modified the first layer of the ResNet model from a 2D CNN to a 3D CNN. This front-end module is used to process sequential frame image sequences. For the second method,



Afouras et al. [21] added a 3D CNN before the 2D CNN. Then, the model used standard ResNet as feature extraction. The model's speech content is separated from background noise to enhance audiovisual speech. However, this method results in a large number of parameter calculations. Xu et al. [22] introduced Pseudo-3D residual convolution (P3D) to extract visual features. The front-end module replaces the ResNet network with the time convolution (TCN). Short-time Fourier transform (STFT) sampling extracts the model's input audio. Then, the model enters the speech enhancement module. The module put enhanced feature information input into the network. This model further improves the accuracy of the identification task. However, the accuracy of the 3D + 2D CNN method is poor when it is disturbed by adversarial examples. How to guarantee the accuracy while improving model robustness and generalization deserves to be investigated. Our model dramatically alleviates this problem using Mixup training, as detailed in Section 4.

Recently, some researchers proposed end-to-end lip, recognition models. This type of model uses fully connected layers [23,24] or convolutional layers [25,26] to extract features from the lip region. Then, features are input into the recursive neural network, attention mechanism [27], or self-attention sequence [26]. Although all these models can achieve good accuracy, they all suffer from the problem of too many parameters, which leads to the large consumption of GPU resources and a long training time. This paper focuses on solving these problems and gives specific methods.

The current state-of-the-art model was proposed by Martinez [28] et al. Since then, many researchers have made improvements on Martinez's basis to obtain higher model accuracy [20,29]. However, these models focus too much on accuracy and neglect the study of model resistance to interference. Moreover, these models often stack neural networks to improve accuracy, which creates the problem of too many parameters and leads to high GPU resource consumption and extended training time. In this paper, we focus for the first time on the anti-interference ability of lip recognition and try to reduce the consumption of the model.

Section 4 analyzes the robustness and generalization of current state-of-the-art lip recognition models when subjected to adversarial attacks generated by FGSM. Experiments show that the accuracy of the advanced lip recognition models significantly decreases when subjected to adversarial attacks, and we invoke Mixup training to improve the accuracy effectively. Meanwhile, in terms of reducing model parameters and resource consumption, we propose the new InvNet-18 network, which reduces 32% of parameters and consumes only 1/3 of GPU resources compared with the ResNet-18 network used by the advanced model.

## 3. The Basic Pipeline

The lip recognition model is divided into front-end modules and back-end modules. The front-end module extracts faces from video datasets. Then, the model extracts mouth features from the face. Next, features enter CNN for space-time encoding. The front-end module finally outputs the space-time feature vector of the lip image. The back-end module uses a cyclic neural network for sequence coding. Furthermore, the model uses a classifier classification [14].

This article uses an advanced model as the baseline, as shown in Figure 4. First is the data processing part. The dataset uses LRW and LRW-1000 video datasets. They are processed into picture frames, then cropped out of the picture of lips as the model's input. The data were processed into the B × T × H × W tensor. Each dimension corresponds to batch, picture frame, height, and width (the input picture has a single channel denoting grayscale). These tensors are input into the front-end module. The front-end module's first convolutional layer uses the 3D CNN. The 3D CNN's convolutional kernel is 5 × 7 × 7 (time-domain, length, width). Then, a standard two-dimensional ResNet-18 is introduced after the 3D CNN. All ResNet-18's convolutional kernels are one-dimensional. ResNet-18 only extracts spatial features. To this point, the model will obtain the model T × 512 × 3 × 3 feature sequence. Then, the feature sequence enters spatial global average pooling (GAP,

the feature dimension is 512, and there are T in total). Finally, the front-end module obtains a 512 × T feature sequence vector. In the dataset, each category of words' duration is short. Therefore, the whole network does not use time-domain downsampling to avoid losing sequence motion information.

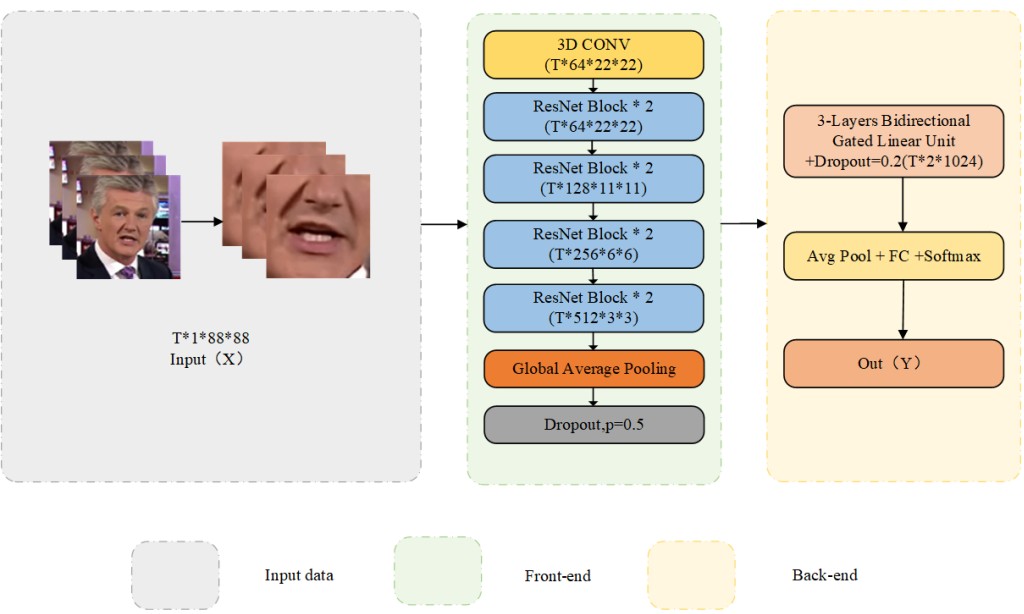

**Figure 4.** Baseline model of this paper.

Afterward, the feature sequence vector enters the back-end module's bi-gated recurrent unit (Bi-GRU). The back-end module is composed of Bi-GRU and a fully connected layer. The back-end module uses the fully connected layer linearly transformed *H*, mapped into a feature vector of dimension *N*. The output of the model $\hat{Y}$ is obtained from the Softmax activation function. The model uses the cross-entropy loss function to calculate the loss, as shown in Equation (1).

$$\text{Loss} = -\frac{1}{N} \sum_{i=1}^{n} Y_i \log \hat{Y}_i \tag{1}$$

## 4. Proposed Methodology

In this paper, an interference-resistant and low-consumption lip recognition model is constructed. Mixup has been shown to improve the robustness and generalization of models in the area of image recognition. However, in the area of lip recognition, no article has detailed the effect of Mixup applied to lip recognition models. The first subsection describes Mixup's rationale for improving model robustness and generalization. Through experiments, it has been demonstrated that Mixup can be used for lip recognition models with good results. We propose the InvNet-18 network in the second subsection to build a low-consumption lip recognition model. This novel network can significantly reduce the model's parameters by the new Involution operator. The resource consumption is reduced by reducing the parameters. The optimized model graph is shown in Figure 5.

### 4.1. Mixup

Mixup is a widely used data enhancement technique introduced by Zhang et al. [6]. It has significantly improved the robustness and generalization of models [6,30]. To address the lack of robustness and generalization of the lip recognition model to adversarial examples, we used the Mixup method on the baseline model and achieved good results.

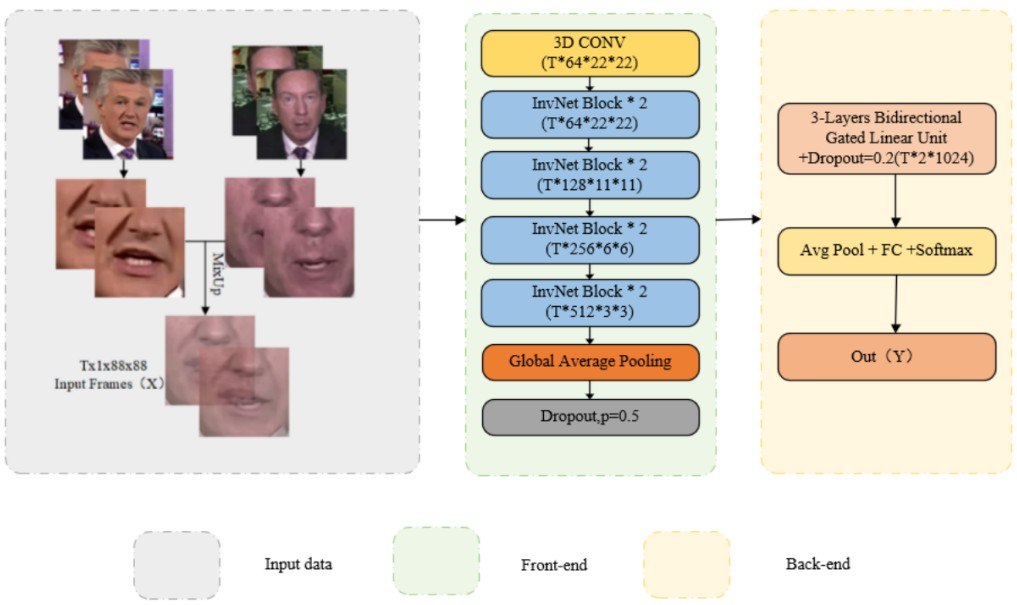

**Figure 5.** The optimized lip language recognition model.

### 4.1.1. Definition of Mixup

In this section, we will briefly review the definition of Mixup.

In the paper by L. Zhang et al. [31], it is demonstrated in detail that Mixup training is equivalent to optimizing a regularized version of the standard empirical loss $L_n^{std}(\theta, S)$. In general, for model classification cases, the output $y_i$ is embedded in the $x_i$ class—that is, using $m$ as the total number of the class and leaving $y_i \in \{0, 1\}^m$ as the binary vector—all items being equal to zero except those corresponding to the $x_i$ class. In particular, if we take $m = 1$, it degrades to binary classification. For regression cases, $y_i$ can be any real number/vector. The Mixup loss is defined as follows:

$$L_n^{\mathrm{mix}}(\theta, S) = \frac{1}{n^2} \sum_{i,j=1}^n \mathbb{E}_{\lambda \sim \mathcal{D}_\lambda} l\big(\theta, \tilde{z}_{ij}(\lambda)\big) \tag{2}$$

where $\mathcal{D}_\lambda$ is the distribution supported on [0, 1]. Throughout the paper, we consider the most commonly used $\mathcal{D}_\lambda$—Beta distribution $\mathrm{Beta}(\alpha, \beta)$ for $\alpha, \beta > 0$.

In this paper, we set the prediction function $f_\theta(x)$ and the target $y$ with a class of loss functions as in Equation (3).

$$\mathcal{L} = \{l(\theta, (x, y)) \mid l(\theta, (x, y)) = h(f_\theta(x)) - y f_\theta(x) \text{ for some function } h\} \tag{3}$$

This function class $\mathcal{L}$ contains many commonly used losses, including loss functions induced by generalized linear models (GLMs), such as cross-entropy and logistic regression for neural networks and linear regression. Mixup training with $\lambda \sim D_\lambda = \mathrm{Beta}(\alpha, \beta)$ introduces a regularized loss function, where the mixture of Beta distributions specifies the weight of each regularization:

$$\tilde{\mathcal{D}}_\lambda = \frac{\alpha}{\alpha + \beta} \mathrm{Beta}(\alpha + 1, \beta) + \frac{\beta}{\alpha + \beta} \mathrm{Beta}(\beta + 1, \alpha) \tag{4}$$

The corresponding Mixup loss $L_n^{\mathrm{mix}}(\theta, S)$, as defined in Equation (2) with $\lambda \sim D_\lambda = \mathrm{Beta}(\alpha, \beta)$, can be rewritten as

$$L_n^{mix}(\theta, S) = L_n^{std}(\theta, S) + \sum_{i=1}^3 \mathcal{R}_i(\theta, S) + \mathbb{E}_{\lambda \sim \tilde{\mathcal{D}}_\lambda}\left[(1 - \lambda)^2 \varphi(1 - \lambda)\right] \tag{5}$$

where $\lim_{a\to 0}\varphi(a)=0$ and

$$\mathcal{R}_1(\theta,S)=\frac{\mathbb{E}_{\lambda\sim\tilde{\mathcal{D}}_\lambda}[1-\lambda]}{n}\sum_{i=1}^{n}\big(h'(f_\theta(x_i))-y_i\big)\nabla f_\theta(x_i)^\top\mathbb{E}_{r_x\sim\mathcal{D}_X}[r_x-x_i] \tag{6}$$

$$\mathcal{R}_2(\theta,S)=\frac{\mathbb{E}_{\lambda\sim\tilde{\mathcal{D}}_\lambda}\big[(1-\lambda)^2\big]}{2n}\sum_{i=1}^{n}h''(f_\theta(x_i))\nabla f_\theta(x_i)^\top\mathbb{E}_{r_x\sim\mathcal{D}_X}\Big[(r_x-x_i)(r_x-x_i)^\top\Big]\nabla f_\theta(x_i) \tag{7}$$

$$\mathcal{R}_3(\theta,S)=\frac{\mathbb{E}_{\lambda\sim\tilde{\mathcal{D}}_\lambda}\big[(1-\lambda)^2\big]}{2n}\sum_{i=1}^{n}\big(h'(f_\theta(x_i))-y_i\big)\mathbb{E}_{r_x\sim\mathcal{D}_X}\Big[(r_x-x_i)\nabla^2 f_\theta(x_i)(r_x-x_i)^\top\Big] \tag{8}$$

This result shows that Mixup is related to regularizing $\nabla f_\theta(x_i)$ and $\nabla^2 f_\theta(x_i)$, which are the first and second directional derivatives with respect to $x_i$. We further denote Equation (2) as

$$\tilde{L}_n^{\mathrm{mix}}(\theta,S):=L_n^{std}(\theta,S)+\sum_{i=1}^{3}\mathcal{R}_i(\theta,S) \tag{9}$$

Equation (9) is equivalent to a regularization version of the optimization standard empirical loss $L_n^{std}(\theta,S)$. This regularization has been shown to improve the robustness and generalization ability of the model greatly. The specific experiments are in the next subsection.

### 4.1.2. Robustness and Generalization

Despite the remarkable success of neural networks in many areas, such as lip recognition [15] and natural language processing [32], it has been noted that neural networks are susceptible to adversarial instances and that predictions are easily flipped by interference [33]. In Goodfellow et al., the authors used the fast gradient sign method (FGSM) to generate adversarial samples, resulting in images of pandas misclassified as gibbons. Although various defense mechanisms have been proposed for negative attacks, they typically sacrifice test accuracy for robustness [34], and many require significant additional computation time. In contrast, Mixup training is somewhat resistant to adversarial examples while improving test accuracy, such as those generated by FGSM [30]. In addition, the corresponding training time is moderate.

As an explanation, this paper compares the Mixup and baseline models for adversarial attacks generated by FGSM. One is a mixed loss model, and the other is a traditional empirical risk minimization (ERM) model. We create FGSM adversarial attack experiments by randomly selecting 2000 test images from the LRW dataset and the LRW-1000 dataset. Since these two datasets are video datasets, we randomly crop each video frame to 88 × 88 sizes. Figure 6a depicts the results of both models. Experiments show that when the attack size is more than five, the model trained by Mixup has higher accuracy than ERM, which is mainly attributed to the excellent robustness of Mixup training.

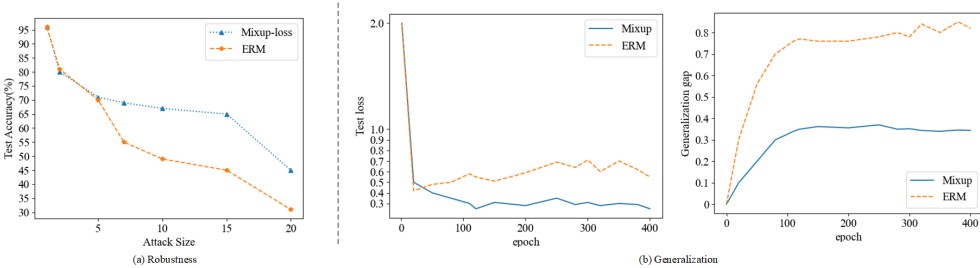

**Figure 6.** Exemplary examples of the impact of Mixup on robustness and generalizability. (**a**) Adversarial robustness of SVHN data under FGSM attack. (**b**) Generalization gap between test and training loss.

Generalization theory has been the focus of deep learning theory [35], and there has been evidence that generalization is an essential measure of whether a deep learning algorithm is good [6]. A model with good generalization will have better test performance. From Figure 6b, With the continuous improvement of the experimental epoch, the test loss and generalization gap of the Mixup method are significantly higher than those of the ERM method. Mixup training yields better test performance than standard ERM methods for Mixup. This result is mainly due to its suitable generalization properties.

After the above experiments, Mixup can significantly improve the robustness and generalization of the lip recognition model when subjected to interference.

### 4.1.3. Mixup and Baseline Model

We further introduced Mixup into the baseline model. The objective is to examine the effect of Mixup on the accuracy of the lip recognition model in the absence of adversarial example interference. The experiment uses full LRW and LRW-1000 as datasets, with training and test sets divided at a ratio of 8:2. Since these two datasets are video datasets, we randomly crop each video frame to 88 × 88. The experiments compared the accuracy of the baseline model with Mixup. The experimental results are shown in Figure 7.

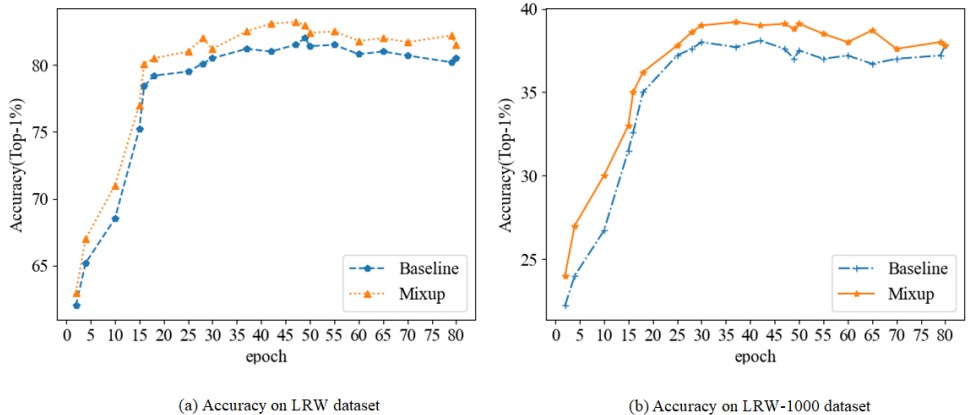

(a) Accuracy on LRW dataset

(b) Accuracy on LRW-1000 dataset

**Figure 7.** TestSet accuracy of Mixup and baseline model. (**a**) TestSet accuracy on LRW dataset. (**b**) TestSet accuracy on LRW-1000 dataset.

As shown in the figure, the model using Mixup is more accurate than the baseline model on both datasets, even when the model is not disturbed by adversarial examples. Mixup helps us construct a lip recognition model that is resistant to interference.

### 4.2. InvNet-18

The involution operator was first proposed by Duo Li et al. [36] as a novel atomic operation for deep neural networks. The involution operator could be leveraged as a fundamental brick to build a new generation of neural networks for visual recognition. This new operator has been shown to reduce the number of parameters and GPU resource consumption of image classification models [36]. We build a new model called InvNet-18 based on ResNet-18 using the involution operator. Then, we demonstrate through detailed experiments that the InvNet-18 model can significantly reduce the model's parameters and save GPU resources.

### 4.2.1. Design of Involution

The conventional convolutional kernels are designed according to spatial invariance and channel specificity. The original convolution kernel shares parameters in space, which leads to limited ability to model space at different spatial locations. It cannot effectively capture long-distance space features. The output channel information of the conventional convolution kernel is synthesized from input channel information. Parameters are not shared, resulting in many parameters and computations. The convolution filters have

information redundancy in different output channels. Using different convolution kernels, each output channel is inefficient. Aiming at two disadvantages of traditional convolution, involution makes improvements. It shares parameters in other groups (the number of groups operating in the convolution), which reduces parameters.

The involution kernels are designed to contain transformations with inverse features in the space and channel domains, which can be represented by $\mathcal{H} \in \mathbb{R}^{H \times W \times K \times K \times G}$. Concretely, an involution kernel $\mathcal{H}_{i,j,\cdot,\cdot,g} \in \mathbb{R}^{K \times K}$, $g = 1, 2, \ldots, G$, is specially tailored for the pixel $\mathbf{X}_{i,j} \in \mathbb{R}^C$ (the subscript of $C$ is omitted for notation brevity), located at the corresponding coordinate $(i, j)$ but shared over the channels. $G$ counts the number of groups sharing the same pairwise kernel. To derive the output feature map of the involution, we perform Multiply–Add operations on the input with such involution kernels, defined as

$$\mathbf{Y}_{i,j,k} = \sum_{(u,v) \in \Delta_K} \mathcal{H}_{i,j,u+\lfloor K/2 \rfloor, v+\lfloor K/2 \rfloor, \lceil kG/C \rceil} \mathbf{X}_{i+u,j+v,k} \tag{10}$$

Unlike convolutional kernels, the shape of the involution kernel $\mathcal{H}$ depends on the condition of the input feature map $X$. Since the model will define the generating involution kernel on the original input tensor, the output kernel will easily be aligned with the input. Define the generating function of the nucleus as $\phi$ and abstract the mapping of the function for each position $(i, j)$ as

$$\mathcal{H}_{i,j} = \phi\left(\mathbf{X}_{\Psi_{i,j}}\right) \tag{11}$$

where $\Psi_{i,j}$ indexes the set of pixels $\mathcal{H}_{i,j}$ is conditioned on. The kernel generation function $\phi : \mathbb{R}^C \longmapsto \mathbb{R}^{K \times K \times G}$ with $\Psi_{i,j} = \{(i, j)\}$ takes the following form:

$$\boldsymbol{H}_{i,j} = \phi(\mathbf{X}_{i,j}) = \mathbf{W}_1 \sigma(\mathbf{W}_0 \mathbf{X}_{i,j}) \tag{12}$$

We refer to Equation (10) with the materialized kernel generation function of Equation (12) as involution hereinafter. The pseudo-code shown in Algorithm 1 delineates the computation flow of involution, which is visualized in Figure 8.

---

**Algorithm 1** Pseudo code of involution in a PyTorch-like style.

---

**Initialization**:
o = nn.AvgPool2d(s, s) if s > 1 else nn.Identity()
reduce = nn.Conv2d(C, C//r, 1)
span = nn.Conv2d(C//r, K*K*G, 1)
unfold = nn.Unfold(k, dilation, padding, s)
**Forward Pass**:
x_unfolded = unfold(x), *B, C*K*K. H*W*
×_unfolded = x_unfolded.view(B, G, C//G, K*K, H, W)
**Kernel Generation, Equation (12)**:
kernel = span(reduce(o(x))), *B, K*K*G, H, W*
kernel = kernel.view(B, G, K*K, H, W).unsqueeze(2)
**Multiply-Add Operation, Equation (10)**:
out = mul(kernel, x_unfolded).sum(dim = 3), *B, G, C/G, H, W*
out = out.view(B, C, H, W)
return out
*B: batch size, H: height, W: width*
*C: channel number, G: group number*
*K: kernel size, s: stride, r: reduction ratio*

---

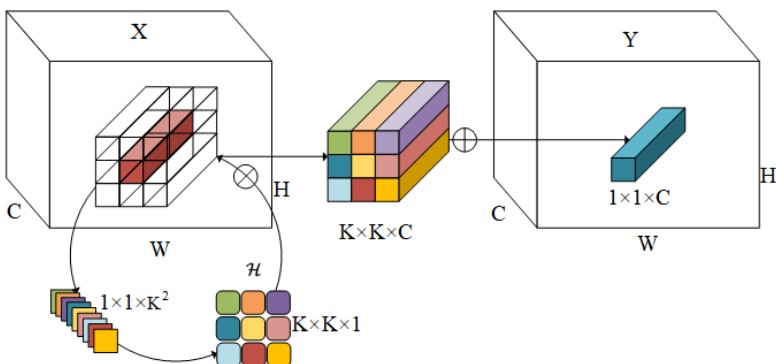

**Figure 8.** Schematic illustration of involution. The involution kernel $\mathcal{H}_{i,j} \in \mathbb{R}^{K \times K \times 1}$ ($G = 1$ in this example for ease of demonstration) is yielded from the function $\phi$ conditioned on a single pixel at $(i, j)$, followed by a channel-to-space rearrangement. The Multiply–Add operation of involution is decomposed into two steps, with $\otimes$ indicating multiplication broadcast across $C$ channels and $\otimes$ indicating summation aggregated within the $KK$ spatial neighborhood.

For building the entire network with involution, we mirror the design of ResNet by stacking residual blocks. We replace the pair fitting in the stem and trunk of ResNet with 3 × 3 convolution at all bottleneck locations but retain all 1 × 1 convolutions for channel projection and fusion. These carefully redesigned networks are called InvNet-18. InvNet-18 consists of 7 ResBlock (as in Figure 9a) and 3 ResDown (as in Figure 9b). The specific parameters are shown in Table 1. Due to the processing of a 5-dimensional tensor, the storage space is ample and categories are few. So, the channels on all layers are down an order of magnitude.

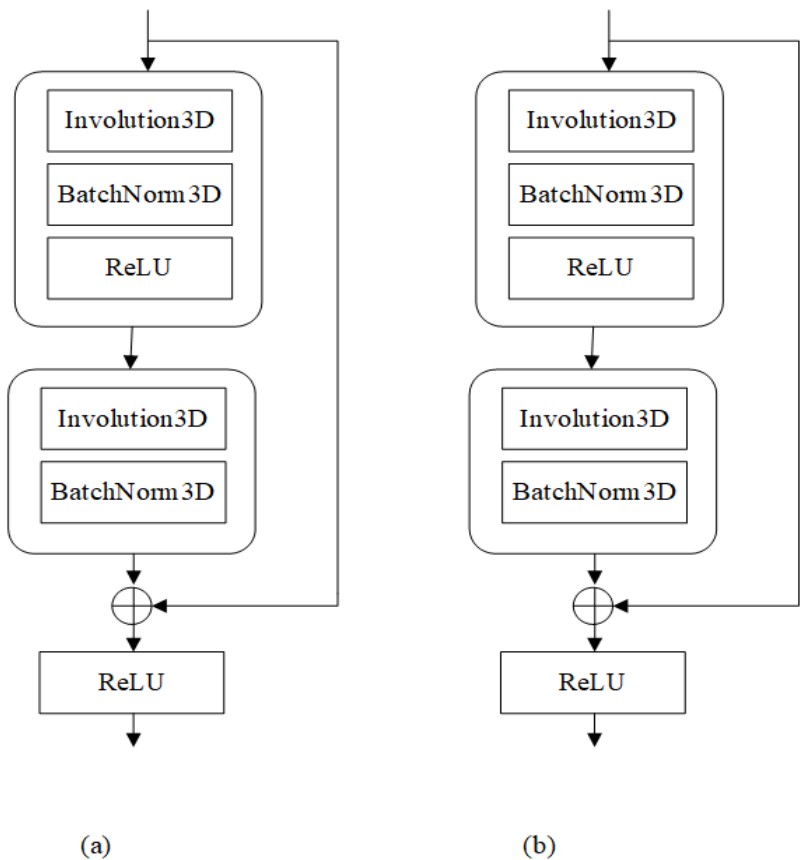

**Figure 9.** 3D ResNet-18 network structure. (**a**) ResBlock module. (**b**) ResDown module.

**Table 1.** 3D ResNet-18 network structure. The name of input and output parameters: batch size, channel, picture frame, height, and width.

| Block Name | Conv Channel | Input Size | Output Size |
|---|---|---|---|
| Conv_pre | 1->64 | [B,1,T,88,88] | [B,64,T,44,44] |
| MaxPool | 64->64 | [B,64,T,44,44] | [B,64,T,22,22] |
| ResBlock1 | 64->64 | [B,64,T,22,22] | [B,64,T,22,22] |
| ResBlock2 | 64->64 | [B,64,T,22,22] | [B,64,T,22,22] |
| ResDown1 | 64->128 | [B,64,T,22,22] | [B,64,T,22,22] |
| ResBlock3 | 128->128 | [B,128,T,11,11] | [B,128,T,11,11] |
| ResDown2 | 128->256 | [B,128,T,11,11] | [B,256,T,6,6] |
| ResBlock4 | 256->256 | [B,256,T,6,6] | [B,256,T,6,6] |
| ResDown3 | 256->512 | [B,256,T,6,6] | [B,512,T,3,3] |
| ResBlock5 | 512->512 | [B,512,T,3,3] | [B,512,T,3,3] |
| Average pool | 512->512 | [B,512,T,3,3] | [B,512,T,1,1] |

4.2.2. The Experimental Effect of Involution

To reflect the actual runtime, we measured the inferred times for different architectures and compared the performance for a single image of shape 224 × 224. We report the runtimes on GPU/CPU in Table 2, where InvNet-18 runs faster in depth at the same accuracy.

**Table 2.** Runtime analysis for representative networks. The speed benchmark is on a single NVIDIA RTX 3070 GPU and Intel® Xeon® CPU E5-2698 v4@2.20 GHz.

| Architecture | GPU Time (ms) | CPU Time (ms) | TOP-1 Acc. (%) |
|---|---|---|---|
| ResNet-50 [37] | 11.4 | 895.4 | 76.8 |
| ResNet-101 [37] | 18.9 | 967.4 | 78.5 |
| SAN19 [38] | 33.2 | N/A | 77.4 |
| Axial ResNet-S [39] | 35.9 | 377.0 | 78.1 |
| InvNet-18 | 11.2 | 156.0 | 77.6 |

In addition, we compared it with other mainstream lip recognition models. The experimental results show that InvNet-18 achieves the highest recognition accuracy whilst with the most parsimonious parameter storage and computational budget. The experimental results are shown in Table 3.

**Table 3.** Comparison results of InvNet-18 and other models.

| Front-End | Back-End | #Params (M) | LRW (%) | LRW-1000 (%) |
|---|---|---|---|---|
| VGGM | N/A | 11.6 | 61.1% | 25.7% |
| ResNet-18 | Bi-GRU | 10.6 | 83.0% | 38.2% |
| ResNet-34 | Bi-LSTM | 19.6 | 83.5% | N/A |
| InvNet-18 | Bi-GRU | 7.2 | 84.5% | 41.6% |

Observed in the above experiments, InvNet-18 can reduce the parameters of the model as well as the computational resources while maintaining accuracy. InvNet-18 helps us to build a low-consumption lip recognition model.

*4.3. The Final Model*

In conclusion, the lip recognition model in this paper can be composed of several steps, as shown in Figure 10.

1. The LRW and LRW-1000 video datasets were decomposed into picture frames and the 88 × 88 size lip image was cropped from them.
2. The front-end module includes 3D Conv, InvNet-18, and GAP to obtain a 512-dimensional time feature sequence and use Mixup training for data enhancement. Mixup enables lip recognition models with high resistance to interference. The InvNet-18 network gives low

consumption to lip recognition models. The combination of the two is suitable for constructing an interference-resistant and low-consumption lip recognition model with an accuracy comparable with the current state-of-the-art results.

3. The back-end module includes Bi-GRU and the full connection layer, resulting in loss and classification results.

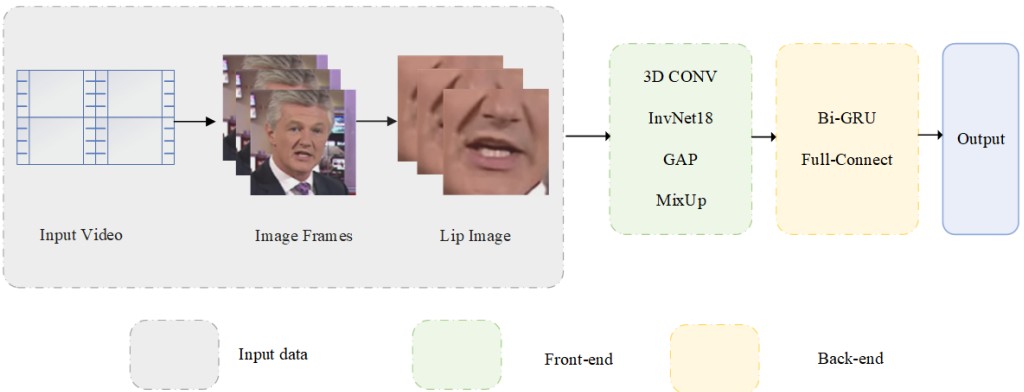

**Figure 10.** The process of lip language recognition in this paper.

## 5. Experimental Result

### 5.1. DataSet

The datasets were selected from the world's largest English lip-reading dataset LRW and the Chinese lip-reading dataset LRW-1000. LRW is a challenging dataset with a 500-word count. It consists of short segments (1.16 s) of BBC programs, its main news, and talk programs. There are over 1000 speakers with significant variations in head pose and illumination. LRW-1000 is also a very challenging dataset with 1000-word classes, 718,018 samples, and a total of about 57 h. The dataset collected data from CCTV TV programs, including background noise, lighting conditions, resolution, posture, gender, makeup, and other speaking environments.

### 5.2. Experimental Setting

All experiments in this paper are performed on a single NVIDIA RTX 3070 GPU and an Intel® Xeon® CPU E5-2698 v4. This test evaluates the model's performance in terms of the accuracy of the test set. As long as the type of the maximum probability value is consistent with the actual type of sample studied, it can be considered accurate. The Top-1 accuracy is the ratio of the sample's expected number to be correct to the total sample's number.

Both datasets are divided into training and test sets at the ratio of 8:2. The data preprocessing module is then utilized, clipping the dataset to 88 × 88; then, Mixup is used for data enhancement. In the front-end module, the convolution core size of the InvNet-18 module is (3, 3, 3), including five downsamplings and one GAP. The module's Batch Normalization [24] is used between each layer. Each GRU has 1024 cells in its hidden layer in the back-end module. There are three layers of Bi-GRU. Moreover, the model's loss uses the cross-entropy function. The model uses the Adam optimizer. The learning rate is initialized to 0.001 with a decay of 0.5 times per round.

The experiment is set for 80 epochs, and the model is validated at the end of each epoch. If the validation error stabilizes in 3 consecutive periods, the learning rate decreases to 0.5 times. The minimum learning rate is set to $1 \times 10^{-6}$. All GRU layers and fully connected layers use dropout to reduce overfitting. The accuracy variation of the training and test sets for each epoch is shown in Figure 11.

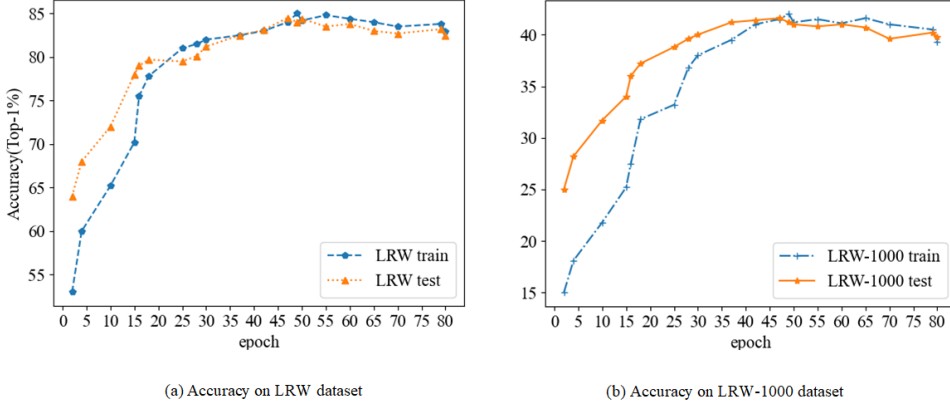

(a) Accuracy on LRW dataset        (b) Accuracy on LRW-1000 dataset

**Figure 11.** Accuracy of the model in this paper.

*5.3. Experimental Results*

The baseline model accuracies for both datasets were 83.0% and 38.2%, respectively, as shown in Table 4. When disturbed by adversarial examples, the accuracy of the baseline model drops by 7.8% and 6.3%, respectively. After the baseline model uses Mixup, the accuracy when bothered by negative examples increases to 85.0% and 40.2%, respectively. The accuracy is higher than the undisturbed lip recognition model in Table 4. This result demonstrates the positive effect of Mixup training on the robustness and generalization of the lip recognition model, which helps us construct an interference-resistant lip recognition model. Tables 2 and 3 have confirmed that the InvNet-18 network can significantly reduce the model's parameters and improve the training compared with other models. This efficient network helps us to build low-consumption lip recognition models. When using Mixup training and InvNet-18 network, the accuracy reaches 85.6% and 41.7%, respectively, which is higher than other lip recognition models in Table 4.

With the above-detailed experiments and explanations, we have proved that Mixup and InvNet-18 can significantly enhance the lip recognition model's various indicators, including the accuracy of the model. The Mixup and InvNet-18 networks can help us build an interference-resistant and low-consumption lip language recognition model.

**Table 4.** Comparison with other models.

| Models | LRW (%) | LRW-1000 (%) |
| :---: | :---: | :---: |
| VGGM | 61.1% | 25.7% |
| D3D | 78.0% | 34.7% |
| GLMIM [40] | 84.4% | 38.7% |
| Baseline model (normal example) | 83.0% | 38.2% |
| Multi-Grained ResNet-18 + Conv BiLSTM | 83.3% | 36.9% |
| ResNet-34 + BiLSTM | 83.5% | 38.2% |
| Two-Stream ResNet-18 + BiLSTM | 84.1% | N/A |
| STCNN + Bi-GRU + Self-Attention [41] | 84.79% | 40.58% |
| Baseline model (adversarial example) | 75.2% | 31.9% |
| Mixup + Baseline model (adversarial example) | 85.0% | 40.2% |
| Mixup + 3D Conv + InvNet-18 + Bi-GRU (normal example) | 85.6% | 41.7% |

## 6. Conclusions

This paper proposes an interference-resistant and low-consumption lip recognition method. No articles analyze the robustness and generalization of lip recognition when subjected to interference. We specifically explore the importance of robustness and generalization to the model and effectively improve the robustness and generalization of the model by using Mixup training. For the current problem of high consumption of lip recognition models, the InvNet-18 network is proposed in this paper.

1. In this paper, we analyze the anti-interference capability of current state-of-the-art lip recognition models and find that they are not robust and generalized enough for adversarial examples, leading to a significant decrease in accuracy for adversarial examples. We experimentally demonstrate that Mixup training can also be applied to lip recognition models to improve their anti-interference ability effectively.

2. Current lip recognition models generally improve the model's accuracy by stacking neural networks, which leads to many parameters and consumes many resources. We propose the InvNet-18 network, which reduces 32% of parameters and consumes only 1/3 of GPU resources compared with the ResNet-18 network used by the advanced model.

In summary, it is proved that Mixup and InvNet-18 can effectively improve the performance of lip recognition. This paper's lip language recognition model is an interference-resistant and low-consumption method.

**Author Contributions:** J.J. collected and analyzed the data, made charts and diagrams, conceived and performed the experiments, and wrote the paper; Z.W. conducted research and investigative processes to collect datasets. L.X., J.D. and M.G. commented, critiqued, and suggested revisions to the first draft of the article. J.H. conceived the structure and provided guidance; J.J. searched the literature and modified the manuscript. All authors have read and agreed to the published version of the manuscript.

**Funding:** This research received no external funding.

**Data Availability Statement:** Not applicable.

**Conflicts of Interest:** The authors declare no conflict of interest.

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
