# Peer review of "An Interference-Resistant and Low-Consumption Lip Recognition Method"

_electronics, doi:10.3390/electronics11193066_

Round 1

Reviewer 1 Report

This paper presents a new technique to improve the anti-interference capability of a lip-movement model of linguistic information and reduce the consumption of training resources. The proposed method introduces the use of Mixup to improve training and proposes the use of a new neural network model based on the Resnet-18 that they call InvNet-18 since it uses an involution operator. The technique presented leads to better results than those obtained with the previously described methods.

Although the paper, in general, is easy to read, the use of English should be improved, both syntactically and semantically, especially in Section 2, lines 74 to 92, where the sentences are written without fluency or continuity, making it necessary to completely rewrite this paragraph.

Some examples are:

Lines 12,15: Replace GUP by GPU.

Line 28: “The Lip recognition model consists of two steps. The first step is to extract the visual features of the lips. The second step is categorization.” ← Please avoid repeating the word step three times.

Line 33: “Although there are corresponding methods [5,6]to solve this problem. However, these methods rely on manual design, and the design process is complex.” ← Please rephrase the text. Perhaps by eliminating the word "however", the sentence would make more sense?

Line 74 a 92: Please correct the paragraph, because the text is not fluent, and there are isolated sentences that are not connected to the others, such as:

"CNN trained with a combination of lip images and phoneme tags."

"To get the feature information.":

Line 125: "But With"

Line 128: Replace "in section 4" by “in Section 4”.

Correct the location of the figures according to the recommendations of the journal, since some are to the right of the text and others in the center.

Equations 7 and 8 should be placed like the others in the center of the text.

Another recommendation is to mention in the abstract the Mixup use, given its importance within the work.

Author Response

Dear Reviewers:

Thank you for your guidance and suggestions, which have benefited us greatly. We have made the following changes based on your suggestions.

1. We have made changes to some grammatical and word errors in the article.
2. In section 2, lines 74 to 92, the sentences are not smooth and coherent, we have rewritten this paragraph.
3. For your question about the position of the images, we used the method suggested by the journal to put some images to the left in a larger scale, because some of them would be unclear if they were placed in the middle.
4. We put equations 7 and 8 in the center of the text.
5. We mention the use of Mixup in the abstract and highlight its importance.

Thank you very much.

Yours sincerely,

Junwei Jia

Reviewer 2 Report

The Authors have been demonstrated through detailed anti-interference experiments that current state-of-the-art lip recognition models have poor robustness and generalization against adversarial examples. The Mixup training has been significantly improved anti-interference ability, robustness, and generalization performance by the Authors. it was experimentally demonstrated that the InvNet-18 network in this paper can effectively reduce model parameters while maintaining accuracy, thus saving GPU resources and reducing training time. The InvNet-18 was an efficien and low-consumption deep neural network. On the datasets LRW[10] and LRW-1000[11], They have been compared the model in this paper with other lip recognition models. It is demonstrated that the model’s accuracy in this paper was comparable to the current state-of-the-art results in the case of interference resistance and low consumption.

My decision is Accept for publication.

Author Response

Dear Reviewers,

We sincerely appreciate your consideration of our manuscripts and look forward to the paper's publication.

Thank you very much.

Yours sincerely,

Junwei Jia

Reviewer 3 Report

1. Word revision:

* In the abstract, there are typos; the GPU word is written as GUP. There are two GUP words in the abstract that need to be corrected.

* In the section 5 (page 14). the word "experimental result" is written as "Experimental ResEults". Please revise the ResEults word.

2. The manuscript is well and clearly presented. However, the comparison of the proposed model with previous studies still needs to be strengthened. We recommend that authors include more previous studies for model comparison (if any). Thus, the robustness of the model could be strongly proven.

3. Figure revision: Figures 7 and 11 appear to be too small. Please increase the quality of those figures.

Author Response

Dear Reviewers:

Thank you for your guidance and suggestions, which have benefited us greatly. We have made the following changes based on your suggestions.

1. We have corrected some typos and grammatical errors in the abstract and the article to make it more fluent.

2. We added a comparison of the model in this paper with previous research models in Section 5 of the article, which strongly demonstrates the robustness of the model in this paper.

3. we enlarged Figures 7 and 11 to make them look clearer.

Thank you very much.

Yours sincerely,

Junwei Jia
